# Improving Sign-Random-Projection via Count Sketch

**Punit Pankaj Dubey**[1]     **Bhisham Dev Verma**[1]     **Rameshwar Pratap**[1]     **Keegan Kang**[2]

[1]Indian Institute of Technology Mandi, H.P., India
[2]Bucknell University, Lewisburg, Pennsylvania, USA

## Abstract

Computing the angular similarity between pairs of vectors is a core part of various machine learning algorithms. The seminal work of Charikar [Charikar, 2002] (*a.k.a.* Sign-Random-Projection (SRP) or SimHash) provides an unbiased estimate for the same. However, SRP suffers from the following limitations: (i) large variance in the similarity estimation, (ii) and high running time while computing the sketch. There are improved variants that address these limitations. However, they are known to improve on only one aspect in their proposal, for *e.g.* [Yu et al., 2014] suggest a faster algorithm, [Ji et al., 2012, Kang and Wong, 2018] provide estimates with a smaller variance. In this work, we propose a sketching algorithm that addresses both aspects in one algorithm – a faster algorithm along with a smaller variance in the similarity estimation. Moreover, our algorithm is space-efficient as well. We present a rigorous theoretical analysis of our proposal and complement it via experiments on synthetic and real-world datasets.

## 1 INTRODUCTION

High-dimensional datasets are ubiquitous in many real-life applications. Performing analytics on such datasets is tedious and, at times impossible due to the *curse of dimensionality*. The dimensionality reduction or sketching algorithms suggest probabilistic algorithmic techniques that compress the high dimensional dataset into low dimensions while preserving pairwise similarity measures such as JL lemma [Johnson and Lindenstrauss, 1983] and its improved variants [Achlioptas, 2001, Li et al., 2006b, Dasgupta et al., 2010, Kane and Nelson, 2014] for real-valued vectors and pairwise euclidean distance. Minhash [Broder et al., 1998]

and its improved variants [Li and König, 2011, Li et al., 2012, Shrivastava, 2017] for sets and pairwise Jaccard similarity. Feature Hashing [Weinberger et al., 2009] and its improved variant [Verma et al., 2022b] preserves the pairwise inner product for real valued vectors. FSketch [Bera et al., 2021] and a duo of Cabin and Cham [Verma et al., 2022a] preserve pairwise hamming distance for categorical vectors. For binary vectors BDR [Pratap et al., 2018a], BCS [Pratap et al., 2018b] and BinSketch [Pratap et al., 2019] preserve the inner product, hamming distance, cosine and Jaccard similarity.

In this work, we focus on the sketching algorithm for real-valued data that approximates pairwise cosine similarity. The seminal work due to Charikar [Charikar, 2002] suggest an algorithm for this task which has been extensively used in applications such as detecting near-duplicates [Manku et al., 2007], Spam-email detection [Ho et al., 2014]. Their algorithm compresses large-dimensional datasets into low-dimensional binary vectors such that the Hamming distance between the sketched vectors gives an unbiased estimate of the pairwise cosine similarity. Let $\vec{a}, \vec{b} \in \mathbb{R}^D$ such that the angle between them is $\theta_{(\vec{a},\vec{b})}$, and let $\mathcal{H} = \{\xi^{(i)}(\cdot)\}_{i \geq 1}$ denote the family of hash function stated as follows:

$$\xi^{(i)}(\vec{a}) = \begin{cases} 1, & \text{if } \langle \vec{a}, \vec{r_i} \rangle \geq 0. \\ 0, & \text{otherwise,} \end{cases} \qquad (1)$$

where $\vec{r_i} = \langle r_{i1}, \ldots, r_{ij}, \ldots, r_{iD} \rangle \in \mathbb{R}^D$ such that $r_{ij} \sim \mathcal{N}(0, 1)$. Repeating the step stated in Equation (1) $K$ times, and concatenating the corresponding hash values gives a $K$ dimensional binary vector corresponding to the input vector. Let $X$ be the estimator random variable for the estimate of cosine similarity by SRP defined as:

$$X = \frac{\pi}{K} \sum_{i=1}^{K} X^{(i)}, \text{where } X^{(i)} = \mathbb{1}_{\xi^{(i)}(\vec{a}) \neq \xi^{(i)}(\vec{b})}.$$
$$\mathbb{E}[X] = \theta_{(\vec{a},\vec{b})}.$$

*Accepted for the 38th Conference on Uncertainty in Artificial Intelligence* (UAI 2022).

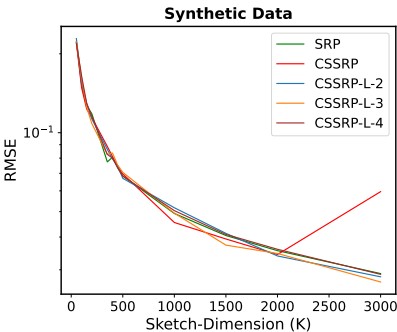

Figure 1: Comparison based on RMSE for angular similarity estimation on a pair of points. Original dimension of points is $10^4$. A smaller RMSE indicates better performance.

$$\text{Var}[X] = \frac{\theta_{(\vec{a},\vec{b})}\left(\pi - \theta_{(\vec{a},\vec{b})}\right)}{K}. \quad (2)$$

SRP can also be seen as a multiplication of the projection matrix (of dimension $K \times D$) with input vectors. Thus, the running time and space required (by the projection matrix) to compute the sketch per data point is $O(DK)$. This highlights the following limitations of SRP:

(i) higher running time and space requirement, especially when the data dimension $D$ is large, and (ii) high variance, when sketch dimension $K$ is smaller, and also when pairwise angle of data points is close to $\pi/2$.

**Previously known improved variants of** SRP**:** There are several results that address some of these limitations of SRP. The result of [Yu et al., 2014, 2018] a.k.a. CBE gives a faster and space efficient algorithm for the task, but its variance remains the same as of SRP, whereas method proposed by [Kang and Wong, 2018] (MLE) and [Ji et al., 2012] (SuperBit) reduces the variance but at the cost of higher running time than SRP. We present an elaborated discussion in Section 2.

To the best of our knowledge, there is no work that addresses all the limitations of SRP in the same sketching algorithm. In this work, we propose one such algorithm. Our key contributions are summarized as follows:

• Our first algorithm is COUNT-SKETCH SIGN-RANDOM-PROJECTION (CSSRP) that compresses high dimensional points into low dimensional binary vectors and closely approximates pairwise cosine similarity, offer a faster running time and simultaneously provide a significantly smaller variance than SRP. At a high level, the CSSRP is inspired from COUNT SKETCH [Charikar et al., 2004a] algorithm, where we first apply the COUNT SKETCH algorithm on the input vectors and then compute the sign of the resultant sketch vector. The similarity estimation step remains exactly the same as SRP (Definition 2). Note that our algorithm can be seen as projecting the input vector on a $K \times D$ projection matrix (whose each column

has exactly one non-zero entry at the index randomly sampled from $\{1, \ldots, K\}$, and takes value between $\{\pm 1\}$ with probability $1/2$), and computing the sign of the resultant $K$-dimensional vector. However, the mentioned improvement of CSSRP holds when the sketch dimension $K = o(D)$ (Theorems 5, 6).

• To alleviate the limitation of CSSRP mentioned above, we propose another sketching algorithm, namely - COUNT SKETCH SIGNED RANDOM PROJECTION-L (CSSRP − L) (Definition 8). The basic difference CSSRP − L and CSSRP is in the process of generating the random projection matrix - each column of the projection matrix for CSSRP − L has exactly $l$ non-zero values (randomly sampled from $\{\pm 1\}$ with probability $1/2$) at randomly chosen positions, where $l \ll K$. CSSRP − L offers significant variance reduction even for large $K$ where $K = o(lD)$ (Theorems 9, 10). We summarise a quick comparison between CSSRP and CSSRP − L using the standard RMSE metric in Figure 1. It is evident that for small values of $K$, CSSRP has a smaller RMSE. However, at higher values of $K$, its RMSE starts increasing, which gets settled by CSSRP − L, even for very small values of $l$ say $2, 3$. Furthermore, both proposals are space-efficient and require $O(D)$ and $O(lD)$ space for projection matrices, for CSSRP and CSSRP − L, respectively, which is significantly less than that required by most baseline algorithms.

• We present our theoretical analysis in Section 4, and complement it via experiments (in Section 5) on synthetic and real-world datasets, on the metrics such as running time, similarity search, and variance analysis via box-plot. We observed a significant speedup (upto $3896\times$) in running time, while simultaneously offering a better performance on the remaining experiments. Our observation is that for small values of the sketch dimension $K$, CSSRP offers both significant speedup and smaller variance, whereas for large values of $K$ CSSRP − L performs similarly even for small values of $l = \{2, 3, 5\}$. We summarise a tabular comparison among the baselines on asymptotic sketching time, space complexity, and variance in Table 1.

## 2 RELATED WORKS

Our work focuses on computing fast and accurate pairwise cosine similarity between input vectors, which has been extensively studied; we summarized some of the related works below:

**CBE:** [Yu et al., 2014] proposed a faster algorithm to compute pairwise cosine similarity. Their algorithm employs a special kind of matrix called circulant matrix, which consists of a random vector $\vec{r} = (r_0, \ldots, r_i, \ldots, r_D)$, where $r_i \in \mathcal{N}(0, 1)$, and $d - 1$ vectors obtained via applying circular shift in $\vec{r}$. Their projection matrix is the matrix obtained

| Algorithm | Sketching Time | Space Complexity | Variance |
|---|---|---|---|
| SRP [Charikar, 2002] | $O(DK)$ | $O(DK)$ | $\theta(\pi-\theta)/K$ |
| CBE [Yu et al., 2014] | $O(D\log D)$ | $O(D)$ | $\theta(\pi-\theta)/K$ |
| SuperBit [Ji et al., 2012] | $O(DK^2)$ | $O(DK)$ | $(\pi^2/K^2)\cdot(K(\theta/\pi)+K(K-1)(\theta/\pi)\times p_{21})-\theta^2$ |
| MLE [Kang and Wong, 2018] | $O(DK)$ | $O(DK)$ | $(2\pi/K)\cdot\left(\frac{1}{\theta+\theta_{\bar{x},\bar{e}}-\theta_{\bar{y},\bar{e}}}+\frac{1}{\theta_{\bar{x},\bar{e}}+\theta_{\bar{y},\bar{e}}-\theta}+\frac{1}{2\pi-\theta_{\bar{x},\bar{e}}-\theta_{\bar{y},\bar{e}}-\theta}+\frac{1}{\theta+\theta_{\bar{y},\bar{e}}-\theta_{\bar{x},\bar{e}}}\right)$ |
| CSSRP (this work) | $O(D)$ | $O(D)$ | $(\pi^2/K^2)\cdot(K(\theta/\pi)+K(K-1)(\theta/\pi)\times\eta)-\theta^2$ |
| CSSRP $-$ L (this work) | $O(lD)$ | $O(lD)$ | $(\pi^2/K^2)\cdot(K(\theta/\pi)+K(K-1)(\theta/\pi)\times\eta_l)-\theta^2$ |

Table 1: Comparison among the baselines on asymptotic sketching time, space complexity, and variance. [Kang and Wong, 2018] conditioned the estimate on a weighted vector $\vec{e}$, hence variance includes the angle formed between the vector pairs and $\vec{e}$. Note that for $\vec{a}, \vec{b} \in \mathbb{R}^D$, in SuperBit, $p_{21}$ is defined as $\Pr[\xi^{(k_2)}(\vec{a}) \neq \xi^{(k_2)}(\vec{b})|\xi^{(k_1)}(\vec{a}) \neq \xi^{(k_1)}(\vec{b})]$, where $\xi^{(k)}(\cdot)$ is the hash function used in SRP (Equation (1)) *s.t.* the rows of the matrix $R$ are orthonormal to each other. $\eta$ and $\eta_l$ are defined in Theorems 6 and 10, respectively.

via multiplication of the circulant matrix and a random diagonal matrix, whose entries are in $\{-1, +1\}$ with probability $1/2$. This projection matrix enables the use of the Fast Fourier transform, which reduces the sketching time to $O(D\log D)$. Moreover, if implemented carefully, the space complexity of the algorithm is $O(D)$. However, its variance remains the same as of the SRP. In comparison, our proposal is not only faster both asymptotically and empirically *w.r.t.* CBE but simultaneously offers a smaller variance.

**SuperBit:** [Ji et al., 2012] proposed an algorithm that offers smaller variance than SRP. Their main idea is to use a projection matrix that consists of orthogonal vectors obtained via the Gram-Schmidt process in $O(DK^2)$ time, which makes its running time high. In comparison, our proposal is much faster both asymptotically and empirically (speedup upto 3896×, see Table 2 and Figure 5), and simultaneously space efficient as well. However, variance expression of both the methods looks similar.

**MLE:** [Kang and Wong, 2018] suggest employing *maximum-likelihood-estimation* technique on top of the sketch obtained from SRP. Inspired by Li et al. [2006a], their techniques include formulating the similarity estimation problem into computing the real roots of a cubic polynomial. In comparison, our proposal is both faster (asymptotically and empirically) as well as space efficient.

In order to understand the comparison among the baselines on their theoretical variances, we plot their respective expressions stated in Table 1. To do so, we generate several data pairs of $10^4$ dimension such that their pairwise angles are between $30^o$ and $150^o$. We summarise it via a scatter plot in Figure 2. It is evident that variances of SRP and CBE remain the highest among all, followed by MLE. Further, the variances of SuperBit, and our proposals CSSRP and CSSRP $-$ L remains the lowest, and are comparable with each other.

Our proposals are based on correlated hash functions. We note that such hash functions have been explored earlier to

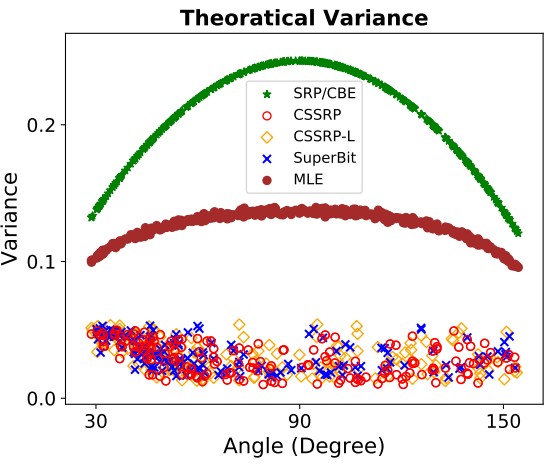

Figure 2: Illustration of theoretical variances of the baselines.

get an accurate estimation for random projection and angular kernel estimation [Choromanski et al., 2017]. Also, the COUNT SKETCH projection matrix used in CSSRP have been used earlier to get a faster algorithm for tasks such as low-Rank approximation and regression [Clarkson and Woodruff, 2013, 2017]. Furthermore, our proposal CSSRP $-$ L uses a projection matrix whose entries are sampled from sparse Bernoulli distribution. We note that such projection matrices have been used in the context of random projection [Li et al., 2006b, Dasgupta et al., 2010, Kane and Nelson, 2014] to get a faster algorithm.

In contrast to the use of the correlated hash functions for variance reduction, statistical techniques such as the control variate trick [Lavenberg and Welch, 1981] and the maximum likelihood estimation method [Murphy, 2012], have been also employed to improve the estimates of different sketching algorithms like AMS sketch [Pratap et al., 2021], Count-Sketch and Count Min Sketch [Pratap and Kulkarni, 2021], Random Projections [Kang et al., 2021], and Feature Hashing [Verma et al., 2022b].

# 3 BACKGROUND

| Notations | |
|---|---|
| $\vec{a}, \vec{b} \in \mathbb{R}^D$ | Input vectors |
| $a_i$ | $i$-th feature of $\vec{a}$ |
| $\vec{\alpha}, \vec{\beta} \in \mathbb{R}^K$ | Sketch vectors |
| $D$ | Original dimension |
| $K$ | Sketch dimension |
| $R$ | Projection matrix |
| $\theta_{(\vec{a},\vec{b})}$ | Angle between $\vec{a}$ and $\vec{b}$ |
| $\vec{r}_k$ | $k$-th row of projection matrix |
| $r_{kj} = s(j)\mathbb{1}_{kj}$ | $(k,j)$-th index of projection matrix |

**Definition 1** (Count-Sketch [Charikar et al., 2004b, Weinberger et al., 2009]). *Let $\vec{\alpha} = (\alpha_1, \ldots, \alpha_k, \ldots, \alpha_K) \in \mathbb{R}^K$ be the sketch of input vector $\vec{a} \in \mathbb{R}^D$, obtained from Count-sketch algorithm. Then, the $k$-th feature of $\vec{\alpha}$*

$$\alpha_k = \sum_{j=1}^{D} a_j s(j)\mathbb{1}_{kj}, \tag{3}$$

*where $s : [D] \mapsto \{-1, +1\}$, and $g : [D] \mapsto [K]$ are hash functions from 2-universal hash families, and $\mathbb{1}_{kj}$ is indicator of the event $g(j) = k$.*

COUNT-SKETCH operation can also be represented as a matrix projection. Let $R$ be a random matrix such that $r_{kj} = s(j) \cdot \mathbb{1}_{kj}$, for all $k \in [K]$, $j \in [D]$.

$$R = \begin{bmatrix} \vec{r}_1 \\ \vdots \\ \vec{r}_k \\ \vdots \\ \vec{r}_K \end{bmatrix}_{K \times D}, \text{ where } \vec{r}_k = (r_{k1}, \ldots, r_{kj}, \ldots, r_{kD}),$$

Therefore, $\vec{\alpha} = R\vec{a}^T$.

# 4 IMPROVING SRP USING COUNT-SKETCH

At a high level our proposal is computing the sketch of input vectors using Count-sketch (see Definition 1) and taking the sign of the resultant vector. We state it as follows.

**Definition 2** (COUNT-SKETCH SIGN-RANDOM-PROJECTION-CSSRP). *We denote our proposal as a hash function $h(\cdot)$ that takes a vector $\vec{a} \in \mathbb{R}^D$ as input, first compress it (say vector $\vec{\alpha} \in \mathbb{R}^K$) using Count-sketch (Definition 1), and then compute the sign of each component of the compressed vector*

$$h(\vec{a}) = \left( h^{(1)}(\vec{a}), \ldots, h^{(k)}(\vec{a}), \ldots, h^{(K)}(\vec{a}) \right).$$

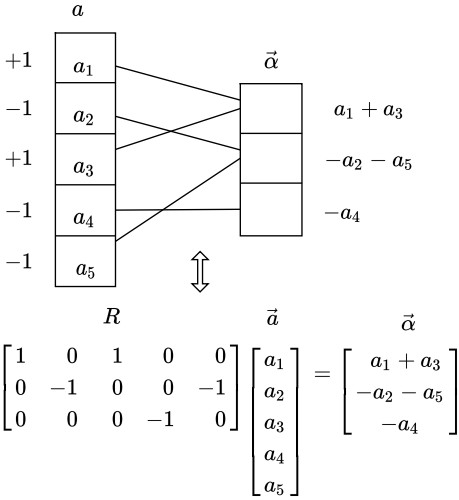

Figure 3: Count-Sketch as matrix projection.

*where, $h^{(k)}(\vec{a}) = sign(\alpha_k)$ and $sign(\alpha_k)$ returns 1 if $\alpha_k \geq 0$, otherwise returns 0.*

In what follows, we prove that our proposal gives an unbiased estimate of the pairwise cosine similarity, further show that the variance of our estimate is smaller than that of SRP.

Our proof techniques relies in showing that the projection matrix (see Figure 3) corresponding to COUNT-SKETCH algorithm, approximates sparse Bernoulli distribution, and further we show that features of the sketch vector obtained from COUNT-SKETCH asymptotically converges to the Gaussian distribution when the sketch dimension $K = o(D)$.

Note that the pairwise angular similarity is only meaningful if all dimensions of the data are more or less equally important; otherwise, the exceptionally large entries will dominate. Therefore, our assumption is that the fourth moment of the input vectors is bounded i.e. $\mathbb{E}[a_i^4] < \infty$, $\mathbb{E}[b_i^4] < \infty$ and $\mathbb{E}[a_i^2 b_i^2] < \infty$, for $\vec{a}, \vec{b} \in \mathbb{R}^D$ (as discussed in Sections 4, 5 of [Li et al., 2006b]). However, the proof of asymptotic normality and analyzing its rate of convergence only require a bounded third moment or even a much weaker condition.

We adopt the following two lemmas from [Li et al., 2006b] to support our proofs. Our all results hold asymptotically as $D \to \infty$.

**Lemma 3.** *[Adapted from Lemma 4 of Li et al. [2006b]] Let $\vec{r} = (r_1, \ldots, r_j, \ldots, r_D) \in \mathbb{R}^D$ s.t.*

$$r_j \sim \begin{cases} 1 & \text{with probability } \frac{1}{2K} \\ 0 & \text{with probability } \frac{K-1}{K} \\ -1 & \text{with probability } \frac{1}{2K} \end{cases} \tag{4}$$

*and $\vec{a} \in \mathbb{R}^D$. Denote $\alpha = \sum_{j=1}^{D} r_j a_j = \langle \vec{r}, \vec{a} \rangle$. Then if $D \to \infty$ and $K = o(D)$, we have $\alpha \xrightarrow{\mathcal{L}} \mathcal{N}\left(0, \frac{||\vec{a}||^2}{K}\right)$ with*

*the rate of convergence*

$$|F_\alpha(y) - \Phi(y)| \leq 0.8\sqrt{K} \frac{\sum_{i=1}^D |a_i|^3}{(\sum_{i=1}^D a_i^2)^{3/2}}$$

$$= 0.8\sqrt{\frac{K}{D}} \frac{\mathbb{E}[|a_i|^3]}{(\mathbb{E}[a_i^2])^{3/2}} \to 0, \quad (5)$$

*where $\overset{\mathcal{L}}{\Rightarrow}$ denotes "convergence in distribution", $F_\alpha(y)$ is the empirical cumulative density function of $\alpha$, and $\Phi(y)$ is the CDF of $\mathcal{N}\left(0, \frac{||\vec{a}||^2}{K}\right)$.*

**Lemma 4.** *Let $\vec{r} \in \mathbb{R}^D$ with the probability distribution in Lemma 3, and $\vec{a}, \vec{b} \in \mathbb{R}^D$. Suppose we denote $\alpha = \sum_{j=1}^D r_j a_j = \langle \vec{r}, \vec{a} \rangle$, and $\beta = \sum_{j=1}^D r_j b_j = \langle \vec{r}, \vec{b} \rangle$. As $D \to \infty$, we have*

$$\sqrt{K} \begin{bmatrix} ||\vec{a}|| & \vec{a}\vec{b}^T \\ \vec{a}\vec{b}^T & ||\vec{b}|| \end{bmatrix}^{-\frac{1}{2}} \begin{pmatrix} \alpha \\ \beta \end{pmatrix} \overset{\mathcal{L}}{\Rightarrow} \mathcal{N}\left(\begin{pmatrix} 0 \\ 0 \end{pmatrix}, \begin{pmatrix} 1 & 0 \\ 0 & 1 \end{pmatrix}\right),$$

*with $\mathbb{E}[||sign(\alpha) - sign(\beta)||_1] = \frac{\theta_{(\vec{a},\vec{b})} \cdot}{\pi}$.*

With the help of Lemmas 3 and 4, in the following we show that CSSRP gives an unbiased estimate of pairwise cosine similarity.

**Theorem 5.** *Let $\vec{a}, \vec{b} \in \mathbb{R}^D$, and $h(\vec{a})$, $h(\vec{b})$ be their $K$-dimensional binary vector obtained via our proposal (Definition 2). If $K = o(D)$, then as $D \to \infty$ we have the following*

$$\mathbb{E}\left[\frac{\pi}{K}||h(\vec{a}) - h(\vec{b})||_1\right] = \theta_{(\vec{a},\vec{b})}. \quad (6)$$

*Proof.* We first consider each row $\vec{r}_k, 1 \leq k \leq K$ of the random matrix in Figure 3. The goal is to find the distribution of each $\vec{r}_k$, and hence compute

$$\mathbb{E}\left[\sum_{k=1}^K |h^{(k)}(\vec{a}) - h^{(k)}(\vec{b})|\right] = \sum_{k=1}^K \mathbb{E}\left[|h^{(k)}(\vec{a}) - h^{(k)}(\vec{b})|\right].$$

Suppose we denote $Z_k := |h^{(k)}(\vec{a}) - h^{(k)}(\vec{b})|$. While each $Z_k$ are not independent due to our construction of $R$, let us briefly consider how each $\vec{r}_k$ is distributed.

When $k = 1$, we have that each entry in $\vec{r}_1$ comes from a Sparse Bernoulli distribution with

$$r_{1j} \sim \begin{cases} 1 & \text{with probability } \frac{1}{2K} \\ 0 & \text{with probability } \frac{K-1}{K} \\ -1 & \text{with probability } \frac{1}{2K} \end{cases}$$

where $\mathbb{E}[r_{1j}] = 0$, with $\text{Var}[r_{1j}] = \frac{1}{K}$. Here, we note that each entry in $\vec{r}_1$ is i.i.d.

We can also compute the moment generating function of each $r_{1j}$ and get

$$\mathbb{E}[e^{sr_{1j}}] = \frac{K-1}{K} + \frac{\exp\{s\} + \exp\{-s\}}{2K}. \quad (7)$$

Now let us consider the case $k = 2$, and compute the moment generating function for each $r_{2j}$. By using the Law of Total Expectation, we have

$$\mathbb{E}[e^{sr_{2j}}] = \mathbb{E}[e^{sr_{2j}} \mid r_{1j} = 0] \mathbb{P}[r_{1j} = 0]$$
$$+ \mathbb{E}[e^{sr_{2j}} \mid r_{1j} = 1] \mathbb{P}[r_{1j} = 1]$$
$$+ \mathbb{E}[e^{sr_{2j}} \mid r_{1j} = -1] \mathbb{P}[r_{1j} = -1].$$
$$= \left(\frac{\exp\{s\} + \exp\{-s\}}{2(K-1)} + \frac{K-2}{K-1}\right) \frac{K-1}{K}$$
$$+ \frac{1}{2K} + \frac{1}{2K}.$$
$$= \frac{\exp\{s\} + \exp\{-s\}}{2K} + \frac{K-2}{K} + \frac{1}{K}.$$
$$= \frac{\exp\{s\} + \exp\{-s\}}{2K} + \frac{K-1}{K}. \quad (8)$$

which is the same moment generating function as the sparse Bernoulli distribution.

Moreover, we also note that each element in $\vec{r}_2$ are i.i.d., i.e. each $r_{2i}$ is independent of $r_{2j}$ (albeit dependent on $r_{1i}$). Now, consider $\vec{r}_k, 2 < k \leq K$, and consider each $r_{kj}$. By Law of Total Expectation, and conditioning on previous vectors:

$$\mathbb{E}[e^{sr_{kj}}] = \mathbb{E}[e^{sr_{kj}} \mid \text{all zeros for } r_{k'j}, k' < k]$$
$$\times \mathbb{P}[\text{all zeros for } r_{k'j}, k' < k]$$
$$+ \mathbb{E}[e^{sr_{kj}} \mid 1 \text{ appears for at most one } r_{k'j}, k' < k]$$
$$\times \mathbb{P}[1 \text{ appears for at most one } r_{k'j}, k' < k]$$
$$+ \mathbb{E}[e^{sr_{kj}} \mid -1 \text{ appears for at most one } r_{k'j}, k' < k]$$
$$\times \mathbb{P}[-1 \text{ appears for at most one } r_{k'j}, k' < k].$$
$$= \left(\frac{K-k}{K-k+1} + \frac{\exp\{s\} + \exp\{-s\}}{2(K-k+1)}\right) \frac{K-k+1}{K}$$
$$+ \frac{k-1}{2K} + \frac{k-1}{2K}.$$
$$= \frac{K-k}{K} + \frac{\exp\{s\} + \exp\{-s\}}{2K} + \frac{k-1}{K}.$$
$$= \frac{\exp\{s\} + \exp\{-s\}}{2K} + \frac{K-1}{K}. \quad (9)$$

which gives the same moment generating function as the sparse Bernoulli distribution.

Now, we can use Lemma 3 to show that $\alpha_k = \langle \vec{r}_k, \vec{a} \rangle$ and $\beta_k = \langle \vec{r}_k, \vec{b} \rangle$ converge in distribution to $\mathcal{N}\left(0, \frac{||\vec{a}||^2}{K}\right)$ and $\mathcal{N}\left(0, \frac{||\vec{b}||^2}{K}\right)$ respectively as $D$ grows large. Moreover, by Lemma 4, we see that $\mathbb{E}\left[|h^{(k)}(\vec{a}) - h^{(k)}(\vec{b})|\right] = \mathbb{E}[|sign(\alpha_k) - sign(\beta_k)|] = \frac{\theta_{(\vec{a},\vec{b})}}{\pi}$ for each $1 \leq k \leq K$. Hence we must have that $\mathbb{E}\left[\sum_{k=1}^K |h^{(k)}(\vec{a}) - h^{(k)}(\vec{b})|\right] = K\frac{\theta_{(\vec{a},\vec{b})}}{\pi}$, and on rearranging, we have

$$\mathbb{E}\left[\frac{\pi}{K} \sum_{k=1}^K |h^{(k)}(\vec{a}) - h^{(k)}(\vec{b})|\right] = \theta_{(\vec{a},\vec{b})} \quad (10)$$

which is what we wanted to show. □

We give a bound on the variance of CSSRP. We defer its proof to the appendix due to space limit.

**Theorem 6.** *Let $\vec{a}, \vec{b} \in \mathbb{R}^D$, and $h(\vec{a})$, $h(\vec{b})$ be their $K$-dimensional binary vector obtained via our proposal (Definition 2). If $K = o(D)$, then as $D \to \infty$ we have the following*

$$\text{Var}\left[\frac{\pi}{K}||h(\vec{a}) - h(\vec{b})||_1\right]$$
$$= \frac{\pi^2}{K^2}\left(\frac{K\theta_{(\vec{a},\vec{b})}}{\pi} + K(K-1)\frac{\theta_{(\vec{a},\vec{b})}}{\pi} \times \eta\right) - \theta^2_{(\vec{a},\vec{b})}.$$

*where, $k_1 \neq k_2$, $k_1, k_2 \in [K]$, and*
$\eta = \text{Pr}\left[\left(h^{(k_2)}(\vec{a}) \neq h^{(k_2)}(\vec{b})\right) \mid \left(h^{(k_1)}(\vec{a}) \neq h^{(k_1)}(\vec{b})\right)\right].$

**Remark 7.** *Recall that the variance of* SRP *is*

$$\frac{\pi^2}{K^2}\left(\frac{K\theta_{(\vec{a},\vec{b})}}{\pi} + K(K-1)\left(\frac{\theta_{(\vec{a},\vec{b})}}{\pi}\right)^2\right) - \theta^2_{(\vec{a},\vec{b})}.$$

*We remark that the variance of* CSSRP *stated in Theorem 6 is smaller than that of* SRP *because $\eta \leq \frac{\theta}{\pi}$. We validate this empirically by plotting $\eta$ for several values of $\theta$ and summarise it in Figure 4. We notice that $\eta$ always remains smaller than $\frac{\theta}{\pi}$, and leads to variance reduction as also supported in Figure 2.*

### 4.1 ANOTHER IMPROVED ESTIMATOR - CSSRP − L:

We note that the stated in Theorems 5 and 6 holds when $K = o(D)$. We wish to show that our results hold for higher values of $K$ as well. Our sketching algorithm CSSRP − L stated below achieves the same.

**Definition 8** (CSSRP − L)**.** *Let $R'$ be a $K \times D$ projection matrix such that each column of $R'$ has exactly $l$ non-zero entries. These $l$ positions are sampled uniformly at random and each of them takes value $\{\pm 1\}$ with probability $1/2$*

$$R' = \begin{bmatrix} \vec{r'_1} \\ \vdots \\ \vec{r'_k} \\ \vdots \\ \vec{r'_K} \end{bmatrix}_{K \times D}. \tag{11}$$

*We denote our proposal* CSSRP − L *as a hash function $h'(\cdot)$ that takes a vector $\vec{a} \in \mathbb{R}^D$ as input, first compress it (say vector $\vec{\alpha'} \in \mathbb{R}^K$) by projecting it on the matrix $R'$ (i.e. $\vec{\alpha'} = R'\vec{a}^T$), and then compute the sign of each component of the compressed vector*

$$h'(\vec{a}) = \left(h'^{(1)}(\vec{a}), \ldots, h'^{(k)}(\vec{a}), \ldots, h'^{(K)}(\vec{a})\right).$$

*where $h'^{(k)}(\vec{a}) = sign(\alpha'_k)$, $sign(\alpha'_k)$ returns $1$ if $\alpha'_k \geq 0$, and $0$ otherwise.*

In the following theorem, we show that our proposal gives an unbiased estimate of pairwise angular similarity. Its proof is built on similar lines to the proof of Theorem 5. We defer it to the appendix.

**Theorem 9.** *Let $\vec{a}, \vec{b} \in \mathbb{R}^D$, and $h'(\vec{a})$, $h'(\vec{b})$ be their $K$-dimensional binary vector obtained via our improved estimator proposal (stated in Definition 8). If $K = o(lD)$, then as $D \to \infty$ we have the following*

$$\mathbb{E}\left[\frac{\pi}{K}||h'(\vec{a}) - h'(\vec{b})||_1\right] = \theta_{(\vec{a},\vec{b})}. \tag{12}$$

We give a bound on the variance of our proposal CSSRP − L estimator, its proof is analogous to that of Theorem 6.

**Theorem 10.** *Let $\vec{a}, \vec{b} \in \mathbb{R}^D$, and $h'(\vec{a})$, $h'(\vec{b})$ be their $K$-dimensional binary vector obtained via our improved estimator (Definition 8). If $K = o(lD)$, then as $D \to \infty$ we have the following*

$$\text{Var}\left[\frac{\pi}{K}||h'(\vec{a}) - h'(\vec{b})||_1\right]$$
$$= \frac{\pi^2}{K^2}\left(\frac{K\theta_{(\vec{a},\vec{b})}}{\pi} + K(K-1)\frac{\theta_{(\vec{a},\vec{b})}}{\pi} \times \eta_l\right) - \theta^2_{(\vec{a},\vec{b})}$$

*where, $k_1 \neq k_2$, $k_1, k_2 \in [K]$, and*
$\eta_l = \text{Pr}\left[\left(h'^{(k_2)}(\vec{a}) \neq h'^{(k_2)}(\vec{b})\right) \mid \left(h'^{(k_1)}(\vec{a}) \neq h'^{(k_1)}(\vec{b})\right)\right].$

**Remark 11.** *Similar to Remark 7, the variance of* CSSRP − L *(Theorem 10) is smaller than that of* SRP *as $\eta_l \leq \frac{\theta}{\pi}$. Its numerical simulation is mentioned in Figure 4. Further, when $l$ is equal to $K$, then rows of the matrix $R'$ defined in Equation (11) become independent, and our proposal* CSSRP − L *(Definition 8) becomes exactly similar to* SRP.

## 5 EXPERIMENTS

**Hardware description:** We conducted our experiments on a machine with the following configuration CPU: Intel(R) Core(TM) i7-7700HQ CPU @ 2.80GHz (8 CPUs); Memory: 8GB; OS: Window 10; Model: MSI GL62M 7RDX.

We use synthetic and real-world dataset for our experiments. In the synthetic dataset, the value of each feature is randomly sampled from $[0, 1]$. Description of real-world dataset is summarized in Table 3.

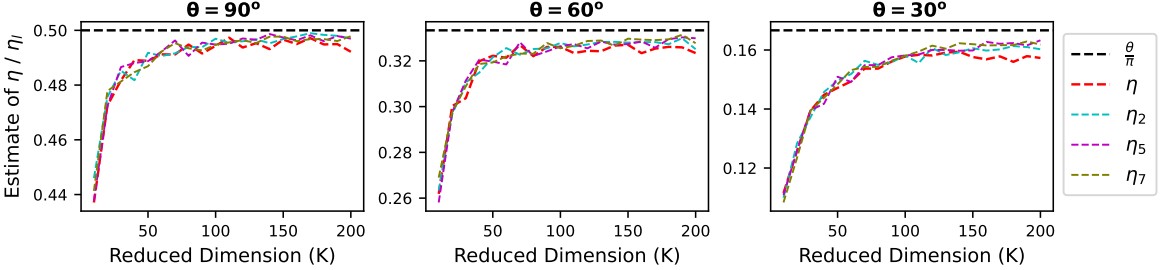

Figure 4: Empirical estimation of $\eta$ and $\eta_l$ via synthetically generated data points for various pairwise angles $\theta$, and reduced dimensions $K$.

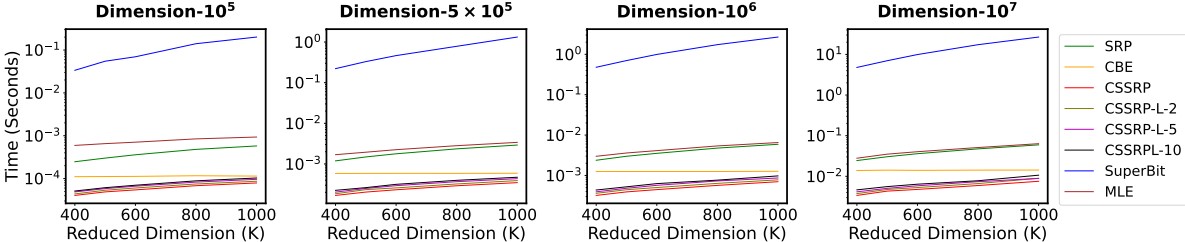

Figure 5: Comparison among the baselines on average running time which consist of both dimensionality reduction time as well as pairwise similarity computation time for a pair. Note that $\mathrm{CSSRP} - \mathrm{L} - 2$ denotes $\mathrm{CSSRP} - \mathrm{L}$ algorithm with $l = 2$, and so on.

Table 3: Description of real-world datasets.

| Dataset | # of points | Dimension |
|---|---|---|
| Gisette [Lichman, 2013] | $13,500$ | $5,000$ |
| Arcene [Lichman, 2013] | $900$ | $10,000$ |
| Gene RNA-Seq [Lichman, 2013] | $801$ | $20,531$ |
| PEMS-SF[Dua and Graff, 2017] | $440$ | $138,672$ |

## 5.1 BASELINES AND END TASKS

We evaluate the performance of our proposals Count-Sketch-Signed Random Projection (CSSRP) and Count-Sketch-Signed Random Projection-L (CSSRP − L) to that of the Signed Signed-random-projection (SRP) [Charikar, 2002], Circulant Binary Embedding (CBE) [Yu et al., 2014], Maximum Likelihood Estimation (MLE) [Kang and Wong, 2018], and Super-Bit LSH (SuperBit) [Ji et al., 2012]. Note that the MLE estimator requires an extra vector for similarity computation, and we use the first principal component vector for the same, as mentioned in [Kang and Wong, 2018]. We use the following metrics for evaluations: (i) running time to generate the sketch, (ii) variance analysis via box-plot, (iii) similarity search.

## 5.2 RUNNING TIME:

**Experimental setting:** We aim to compare the running time of all the baselines. To do so, we generate high dimensional synthetic datasets of dimensions ranging from $10^5$ to $10^7$.

We compress the datasets for different values of reduced dimension using various baselines, and record the sum of sketching time and pairwise similarity computation time. Note that the sketching approach of MLE remains same as that of SRP, however its similarity estimation step is different, and involves computing the root of a cubic polynomial. Therefore to have a fair comparison among all the baselines, we included both sketching time as well as similarity computation time. We compute average running time required by a pair of points, over various reduced dimensions, and summarise it in Figure 5. We also note the corresponding speedup obtained via our proposal CSSRP *w.r.t.* baselines, and report it in Table 2.

**Insight:** We observed that CSSRP is much faster than all the baselines, and we observed a significant numerical speedup (upto $3800\times$). We would like to highlight that our CSSRP is also faster (speed up $1.45\times$ to $2\times$) than CBE [Yu et al., 2014], which is a faster variant of SRP. Further, the running time of our other proposal CSSRP − L remains somewhat comparable to CSSRP. Note that the SuperBit method remains the slowest among all the baselines; this is due to the step of generating orthonormal vectors (via *Gram-Schmidt* orthogonalization process) required for the projection matrix.

## 5.3 SIMILARITY SEARCH:

**Experimental setting:** In this experiment, aim is to check if points proximity are maintained after dimensionality re-

| Estimators | Dimension-$10^5$ | Dimension-$5 \times 10^5$ | Dimension-$10^6$ | Dimension-$10^7$ |
|---|---|---|---|---|
| SRP [Charikar et al., 2004a] | $7.32\times$ | $8.41\times$ | $8.44\times$ | $8.65\times$ |
| CBE [Yu et al., 2014] | $1.45\times$ | $1.70\times$ | $1.811\times$ | $2.074\times$ |
| MLE [Kang and Wong, 2018] | $11.80\times$ | $9.79\times$ | $9.32\times$ | $9.24\times$ |
| SuperBit [Ji et al., 2012] | $2541.35\times$ | $3820.91\times$ | $3826.56\times$ | $3896.71\times$ |
| CSSRP $-$ L $-$ 2 (this work) | $1.09\times$ | $1.12\times$ | $1.15\times$ | $1.20\times$ |
| CSSRP $-$ L $-$ 5 (this work) | $1.22\times$ | $1.25\times$ | $1.29\times$ | $1.39\times$ |
| CSSRP $-$ L $-$ 10 (this work) | $1.31\times$ | $1.35\times$ | $1.40\times$ | $1.55\times$ |

Table 2: Numerical speedup of CSSRP *w.r.t.* other baselines on a fixed reduced dimension 1000.

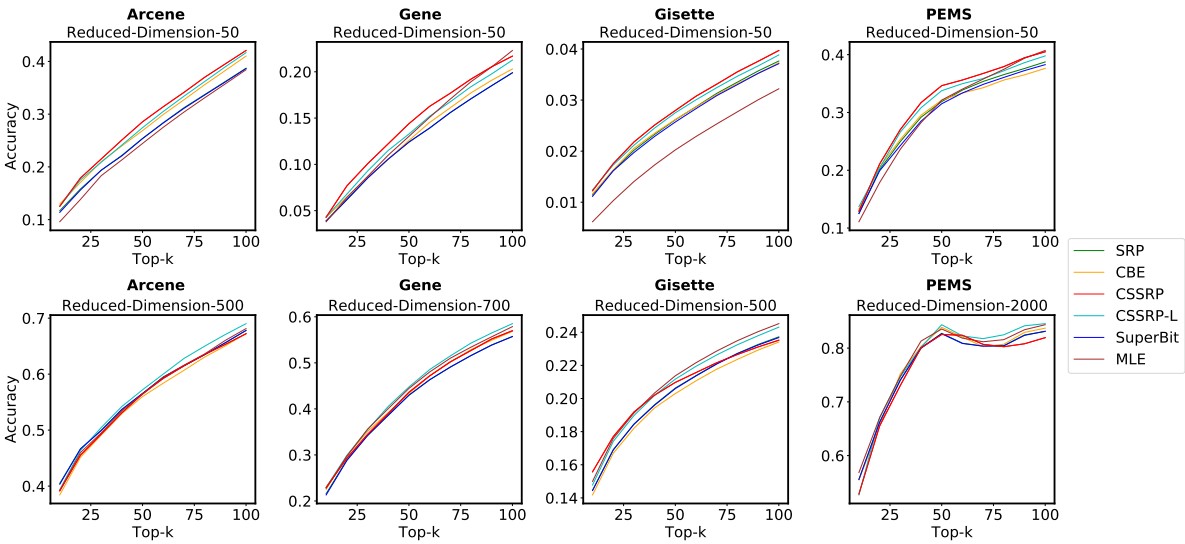

Figure 6: Comparison among the baselines on the task of Top-$k$ similarity search. A higher value of accuracy indicates a better performance.

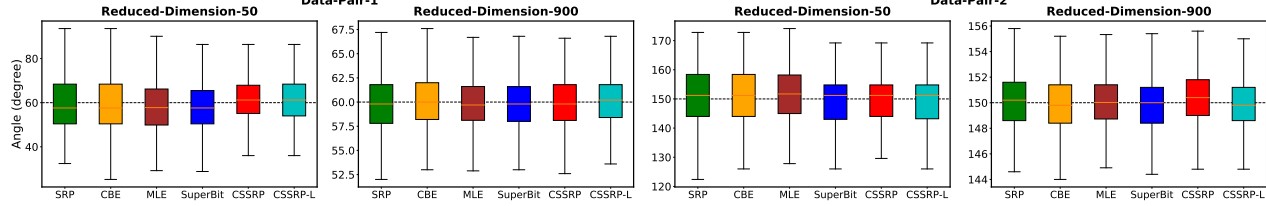

Figure 7: Comparison among baselines on the task of variance analysis via box plot. The sampled pairs are at angles $60°$ and $150°$, respectively. The smaller interquartile range is an indicator of lower variance. The dotted line represents the actual angle in degree.

duction. We discuss our experimental setting as follows. We split the dataset randomly into two parts $90\%$ and $10\%$ – we refer the former as the training partition, while the latter one as the query partition. For each point in the query partition, we record top-$k$ similar points (under cosine similarity) from training partition for the uncompressed datasets. We denote this set by $S$. We compress the dataset (both query and training partition) using several baselines on various reduced dimensions, and record top-$k$ similar points (on the sketch) of the query points from the sketch of the training partition. We denote this set by $S^{'}$. We use two evaluation metrics – *recall*:= $|S \cup S^{'}|/|S|$, and *accuracy*= $|S \cap S^{'}|/|S \cup S^{'}|$.

We compute them for all the points in the query partition, and record their average. We summarize our findings for accuracy in Figure 6 and for recall in appendix.

**Insight:** We observed that at small reduced dimensions (listed in first rows of the respective plots) for both accuracy and recall, our estimator CSSRP estimator performed significantly better than the baselines. However, with the increase of dimension CSSRP performance slightly decreases (listed in second rows of the respective plots), which was circumvented by our other proposal CSSRP $-$ L, whose performance remains at least in the top two.

## 5.4 VARIANCE ANALYSIS VIA BOX-PLOT:

**Experimental setting:** In this experiment, our aim is to compare the variances of the baselines via box-plot. To do so, we generate a synthetic dataset in $10000$ dimension, and randomly sample a pair of points from it. We compress this pair and compute the estimated similarity using all the baselines. We repeat this step $500$ times independently, and use the respective estimate to generate the box plot. We summarise our findings in Figures 7.

**Insight:** We observe that at a small reduced dimension, the variance of our CSSRP estimator is lower than the variance of the other baselines. However, at higher reduced dimension variance of CSSRP is slightly worse than the remaining. This problem is tackled by our other proposal CSSRP − L, which offers smaller variance than the baselines, even at higher values of the reduced dimension.

## 6 CONCLUSION

We consider dimensionality reduction for real-valued data that approximate cosine similarity. The classical algorithm for this task - SRP [Charikar, 2002] suffers from high variance, running time, and space complexity involved in the similarity computation. Popular improvements such as [Kang and Wong, 2018, Yu et al., 2014, Ji et al., 2012] address only one or two aspects of the above. We present algorithms (CSSRP and CSSRP − L) that address all these limitations. When the sketch dimension $K = o(D)$, our proposal CSSRP offers a faster and space-efficient algorithm along with the smaller variance. However, for large $K$, the guarantee of CSSRP does not hold. Our other proposal CSSRP − L, addresses this by offering a faster and space efficient algorithm with smaller variance, when $K$ is large. We give a theoretical analysis of our proposals and complement it via empirical simulations. We notice the speedup of several orders (even with faster variants of SRP [Yu et al., 2014]) and simultaneously accurate performance on end tasks, *w.r.t.* baselines. Finally, we could only empirically show that our proposals have smaller variance *w.r.t.* the baselines. Giving its mathematical proof still remains an open question of the work.

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
