# OpenReview forum: "Improving Sign-Random-Projection via Count Sketch"
_auai.org/UAI/2022/Conference — UAI 2022 Poster_

### Official Review · Reviewer_fkkh · 2022-04-12

**Q2(1) Originality/Novelty:** 3
**Q2(2) Significance/Impact:** 3
**Q2(3) Correctness/Technical Quality:** 2
**Q2(6) Clarity Of Writing:** 2
**Q6 Overall Score:** 6
**Q8 Confidence In Your Score:** 2

**Q1 Summary And Contributions:**

The paper gives an algorithm to compute angular similarity between pairs of vectors. It proposes a sketching algorithm that is faster and has a smaller variance in the similarity estimation than state-of-the-art.

**Q2 Assessment Of The Paper:**

More detailed information regarding each of these aspects is given below:

**Q2(4) Quality Of Experiments (Optional):**

3: Good: The experimental evaluation is adequate, and the results convincingly support the main claims.

**Q2(5) Reproducibility:**

3: Good: Key resources (e.g., proofs, code, data) are available and key details (e.g., proofs, experimental setup) are sufficiently well-described for competent researchers to confidently reproduce the main results.

**Q3 Main Strengths:**

Main strength: An algorithm that is clearly better than the state-of-the-art baselines in a well-defined problem (similarity estimation).

Both theoretical and experimental evidence, experiments are very strong.

**Q4 Main Weakness:**

The writing is not great at times. I find it bit hard to read the proofs of the Appendix. For example, you should write above or below the convergence arrows that D goes to infty or something similar, otherwise it is very difficult to read those results. There are some substeps missing, for example I cannot see how (47) follows from (46).

**Q5 Detailed Comments To The Authors:**

Please try to polish the paper and improve the writing as much as you can in case of acceptance. Please polish also the mathematics of the appendix.

**Q7 Justification For Your Score:**

I think this looks like a good fit for this venue: a state-of-the-art methods that beats the baselines in a well-defined ML task. Has both theoretical analysis and strong experimental evidence.

**Q9 Complying With Reviewing Instructions:**

1: Yes.

---

### Official Review · Reviewer_ASSs · 2022-04-13

**Q2(1) Originality/Novelty:** 3
**Q2(2) Significance/Impact:** 3
**Q2(3) Correctness/Technical Quality:** 3
**Q2(6) Clarity Of Writing:** 4
**Q6 Overall Score:** 7
**Q8 Confidence In Your Score:** 4

**Q1 Summary And Contributions:**

The authors consider two variants on Charikar's Sign Random Projection (SRP) algorithm in which (a) the projection costs less per point to compute, and (b) the variance of the estimator for the cosine distance is significantly smaller than for SRP.  The results are backed by both theory and some experimental evidence.

**Q2 Assessment Of The Paper:**

More detailed information regarding each of these aspects is given below:

**Q2(4) Quality Of Experiments (Optional):**

3: Good: The experimental evaluation is adequate, and the results convincingly support the main claims.

**Q2(5) Reproducibility:**

3: Good: Key resources (e.g., proofs, code, data) are available and key details (e.g., proofs, experimental setup) are sufficiently well-described for competent researchers to confidently reproduce the main results.

**Q3 Main Strengths:**

The idea of replacing random projections with CountSketch (or the l-nonzero-per-column variant) is easy to implement and gives remarkably good results.  The paper is quite well explained.

**Q4 Main Weakness:**

The analysis of the (asymptotic) variance depends on the mysterious eta_l.  The bounds given show that the variance of the estimator should be lower than SRP, but the experiments seem to show that the variance is *much* lower, and the theory for why is not yet complete.  This is also most significant in settings with relatively few queries relative to the per-point computations (and relatively few points relative to the setup costs).

**Q5 Detailed Comments To The Authors:**

The cost to compute a query is about the same between all the algorithms considered.  If there are a large number of queries, the per-query cost may dominate.  This might matter if computing similarities between every pair for a large number of points, for example.  Similarly, the cost of SuperBit (as opposed to the SRP algorithm) is purely in the setup cost; it costs DK^2 to orthogonalize the random vectors, but that is a fast DK^2 if done properly (using algorithms like those found in LAPACK and BLAS).

Similarly, I wonder about the implementation details hidden behind the relative cost numbers.  Using a tuned BLAS and LAPACK to do the QR factorization and to compute the hashes (which should be done for several points at a time in order to take full advantage of level 3 BLAS), I would expect these algorithms to run much faster for data sets of the size reported.  I don't think what is stated is dishonest or anything like that, but to give proper context for the performance numbers, I think it would be worthwhile commenting explicitly that this is done in Python and the implementation is serial (so the 8 cores reported in the hardware aren't actually that significant).

**Q7 Justification For Your Score:**

This seems like a good piece of work in the sketching space, and it's nice that it improves both variance and speed simultaneously.  It would be nicer if the variance was a little more tightly bounded, but I would be satisfied to see that as a subject for future work.

**Q9 Complying With Reviewing Instructions:**

1: Yes.

---

### Official Review · Reviewer_JpJi · 2022-04-13

**Q2(1) Originality/Novelty:** 3
**Q2(2) Significance/Impact:** 2
**Q2(3) Correctness/Technical Quality:** 3
**Q2(6) Clarity Of Writing:** 3
**Q6 Overall Score:** 7
**Q8 Confidence In Your Score:** 3

**Q1 Summary And Contributions:**

This paper proposes two improvements to the sign-random-projection (SRP) sketching algorithm that approximates pairwise cosine similarity. The improvements both lower the variance as well as improve the run time. 1) first applying count-sketch. This has a limitation K=o(D). 2) to deal with limitation: modify count-sketch so that each input dimension is mapped to multiple output dimensions by the hash function, instead of one. This weakens the aformentioned limitation to K=o(lD).

**Q2 Assessment Of The Paper:**

More detailed information regarding each of these aspects is given below:

**Q2(4) Quality Of Experiments (Optional):**

3: Good: The experimental evaluation is adequate, and the results convincingly support the main claims.

**Q2(5) Reproducibility:**

3: Good: Key resources (e.g., proofs, code, data) are available and key details (e.g., proofs, experimental setup) are sufficiently well-described for competent researchers to confidently reproduce the main results.

**Q3 Main Strengths:**

The technique seems useful as it can speedup machine learning with high dimensional data.
The work seems technically sound
the experiments are convicing

**Q4 Main Weakness:**

The writing could be improved, now it is very heavy on the mathematics. While the mathematics are important, a bit more intuition could be helpful. E.g. say in words what effect is expected from the second improvement, and why it can be achieved with the improvement. Now it's mostly left implicit in the formulas.

**Q5 Detailed Comments To The Authors:**

My main concern is that the intuitive reason why the two improvements should work are left implicit. Right now, this is all hidden in mathematics. It would be good to get answers to the most important questions in plain english.  Why does applying count sketch first reduce the variance? Why is a minimal number of buckets K necessary for it to work? Why can this be alleviated by having the hashing functions map to multiple buckets?

The introduction does do a very good job at introducing the topic to non-experts. Adding those extra intuitions, would make it a nice paper for me.

**Q7 Justification For Your Score:**

The results are interesting and could be moderately impactful. The work is well substantiated (theoretically and empirically).

**Q9 Complying With Reviewing Instructions:**

1: Yes.

---

### Official Review · Reviewer_RW1T · 2022-04-14

**Q2(1) Originality/Novelty:** 3
**Q2(2) Significance/Impact:** 2
**Q2(3) Correctness/Technical Quality:** 2
**Q2(6) Clarity Of Writing:** 2
**Q6 Overall Score:** 4
**Q8 Confidence In Your Score:** 2

**Q1 Summary And Contributions:**

The authors propose a random projection method that hashes data points to low-dimensional binary vectors such that the cosine similarity between points is represented by the Hamming distance between binary vectors. Theoretical results state that the proposed method is in expectation maintaining the cosine distances and that the variance of the similarity estimators is bounded. Empirical experiments compare the proposed method against 4 competitors on synthetic and real-world data.

**Q2 Assessment Of The Paper:**

More detailed information regarding each of these aspects is given below:

**Q2(4) Quality Of Experiments (Optional):**

2: Fair: The experimental evaluation is weak: important baselines are missing, or the results do not adequately support the main claims.

**Q2(5) Reproducibility:**

1: Poor: Key details (e.g., proof sketches, experimental setup) are incomplete/unclear, or key resources (e.g., proofs, code, data) are unavailable.

**Q3 Main Strengths:**

1) The experiments are well designed, synthetic and real-world data is used to validate the theoretical results
2) If the provided theory actually holds, then this could be very valuable paper with a good contribution
3) Relations to the existing literature is put well into context

**Q4 Main Weakness:**

1) Reproducibility is not given. I could not infer from the paper which hash functions $g$ and $h$ are now actually used. Hence the core method is entirely a black box to me since I don't know how the hash functions are computed. The proofs make use of properties of the hash functions that are not mentioned in the paper and hence the proofs are impossible for me to verify (see minor comments)
2) The notation is unclear/confusing. In Def. 1 the hash functions $g$ and $h$ are mapping from a set of indices to another set of indices, later  $h$ is mapping from the data points to a vector and the index is annotated as superscript.
3) In the plots it's often hard to distinguish between the methods, some of the plots should rather be tables.

**Q5 Detailed Comments To The Authors:**

* Eq. (1): $\mathbb{E}[X]=\theta_{(a,b)}$ is weird. $X$ is a matrix and $\theta$ is a value, this doesn't go together, same with the variance below.
* What the RMSE actually reflects (RMSE of what?) is unclear in Fig.1. Also, it's hard to see a difference between the methods, except for this one kink of CSSRP.
* I would denote the hash functions in Def. 1 as $g(a)$ and $h(a)$, and the value at index $i$ as $g(a)_i$ and $h(a)_i$ and then stick to this nottatioon. Where is the definition oof $g$ and $h$?
* Use  another function name for $h$ in Def. 2, because it's a different function than $h$ in Def. 1
* I don't understand how the conclusion that $r_1$ comes from a sparse Bernoulli distribution comes from when $k=1$ in the Thm. 5 proof.
* Definition 8: $\alpha=R'a^T$ doesn't compute, the transposed has to be removed, same with $\beta$
* I don't understand what the sense is of considering $o(lD)$, what is $l$ supposed to be, why does the expected value not depend on $l$, or why is nothing influenced by $l$.
* Remark 11: Why are the rows of $R'$ independent when $l=K$?
* Table 2: What is the base for which the speedup is computed? SRP is 7.32x faster than what?
* Figure 7: You want to compute the variance and the mean, boxplots are here not really suitable. Denote the values in a table stating mean$\pm$ std.
* Eq. (33): why is $Pr[h(a)_k \neq h(b)_k] = \theta_{(a,b)}/ \pi$?

### After Rebuttal Thoughts
The authors clarified many of my questions but I would have to go in detail through the paper again to properly assess it. In particular, getting into the proofs is really time-consuming. Since I don't have the time to do so, I'll just reduce my confidence score and let the reviewers who understand the paper decide. I'll also upgrade my score since the authors could clarify the questions I had. The rebuttal has shown that often, the information I was missing was hinted at, but it just wasn't clear. Sometimes, a formula or an extra sentence would help to clarify what is now exactly meant. I hope that my questions help tthe authors in clarifying the paper.
I do have two remaining remarks:
* I would really not use row-major format for vectors. I've never ever seen this in any ML paper and it's just really confusing.
* I'm still not convinced about the boxplots in Figure 7. When you want to analyze the variance then a boxplot is not suitable. If you want to show a distribution, then you can use density estimation plots.




**Q7 Justification For Your Score:**

Promising paper which leaves out the key definition of the proposed hash methods and their properties which are necessary for the theoretical results.

**Q9 Complying With Reviewing Instructions:**

1: Yes.

---

### Decision · Program_Chairs · 2022-05-15

**Decision:**

Accept (Poster)

**Comment:**

Meta Review: The topic of the manuscript is fast and accurate approximation of the angular similarity between vectors in R^D. Particularly, the goal of the authors is to mitigate the existing improvements of the SRP (signed random projection) methods which are either fast or have small variance. By relying on the Count Sketch approach they propose two techniques for which both desiderata hold [CSSRP (count-sketch sign-random projection) and CSSRP-L (count sketch signed random projection-L)], with theoretical results showing their unbiased property and variance (Theorem 5-6, Theorem 9-10). The efficiency of the approach is illustrated and compared against 4 baseline methods.

Approximate computation of pairwise similarities is a central topic of machine learning with a large number of successful applications. The authors deliver important tool in this domain which has clear practical interest and backed by performance guarantees.